# Electrofenton with Reticular Vitreous Carbon and Iron Oxide Nanoparticles for Dye Removal: A Preliminary Study

Fernanda L. Rivera, Nieves Menendez [ORCID], Eva Mazarío [ORCID] and Pilar Herrasti *

Department of Applied Physical Chemistry, Faculty of Science, Autonomous University of Madrid, 29049 Madrid, Spain
* Correspondence: pilar.herrasti@uam.es

**Abstract:** In this work, an RVC electrode coated with magnetic iron oxide nanoparticles was used for the degradation of methylene blue as a model dye. The electrofenton process was carried out by the reduction of oxygen dissolved in the electrolyte on the modified RVC electrode to produce hydrogen peroxide. The presence of the magnetite/maghemite nanoparticles in the structure produces the formation of OH. radicals that oxidize methylene blue. The RVC/coated was prepared by two different methodologies: Methodology A: by immersion of the electrode in a solution saturated with magnetite nanoparticles; and Methodology B: by electrochemical synthesis. Scanning electron microscopy, X-ray diffraction and Mössbauer spectroscopy shows a uniform coating of the electrode. The Mössbauer spectroscopy determines the only presence of maghemite using methodology A and the presence of 60% of magnetite and a 40% of maghemite when methodology B was used. The dipping methodology is the one that has provided the best results in the electrofenton degradation of methylene blue, obtaining a 100% removal after 35 min, applying a current of 100 mA in a 20 mg $L^{-1}$ solution of methylene blue, and a concentration of 50 mM sodium sulfate.

**Keywords:** electrofenton; iron oxide nanoparticles; reticular vitreous carbon; degradation; dyes; hydrogen peroxide





## 1. Introduction

Advanced Oxidation Processes (AOPs) are capable of degrading recalcitrant toxic pollutants, which are not possible to eliminate through other treatment methods. These advanced processes are based on the generation of a highly reactive species (hydroxyl radicals, OH.), which oxidize the organic pollutant [1]. The most common AOP is the Fenton reaction; this is based on the reaction of $H_2O_2$ with iron ions. Although the aforementioned process is efficient, some of the main disadvantages of this treatment are the industrial production, storage, transportation of the $H_2O_2$ and the formation of sludge. To avoid these problems, solid supports that include iron or metals compounds in their structure have begun to be used to provide the active oxygen species needed for the Fenton process, this AOP has been called Electrofenton-like. This methodology has advantages over the Fenton process; hydrogen peroxide is produced directly on the electrode surface and the presence of iron ions in solution avoids the formation of sludge and allows the recovery of the catalyst at the end of the treatment. Many studies are being directed towards the immobilization of iron species or insoluble iron oxides.

Hydrogen peroxide can be continuously electrogenerated at the cathode electrode by the reduction oxygen via two electrons (Reaction (1)), follow by further 2 electron reduction to $H_2O$, (Reaction (2)). Oxygen can also be reduced directly to water via four electrons (Reaction (3)), these processes can occur simultaneously, and therefore decrease the concentration of hydrogen peroxide needed. To avoid this, it is necessary to check that

the potential does not exceed the value corresponding to Reaction (1). In addition, the presence of iron ions also favorably catalyzes Reaction (1) instead of the other reactions.

$$O_2(g) + 2H^+ + 2e^- \rightarrow H_2O_2 \; E^0 = 0.695 \text{ vs. SHE} \tag{1}$$

$$H_2O_2 + 2H^+ + 2e^- \rightarrow 2H_2O \; E^0 = 1.776 \text{ vs. SHE} \tag{2}$$

$$O_2(g) + 4H^+ + 4e^- \rightarrow 2H_2O \; E^0 = 1.23 \text{ vs. SHE} \tag{3}$$

The electrodegradation process was first applied to wastewater treatment by Sudoh et al. [2] and popularized by Oturan's and Brilla's groups since the nineties using carbon felt (CF) and gas diffusion electrodes (GDEs), respectively [3,4]. The use of carbon is attractive due to the characteristics of this material: it is cheap, has large surface area, is inert, and presents good conduction of the electrical current. Additionally, carbon materials prefer the two electron process instead the reduction of oxygen to produce $H_2O$ via 4 $e^-$ [5]. The graphite [6], sponges [7], carbon fibers [8] and the reticulated vitreous carbon (RVC) [9], are just some examples that have been used in these processes to produce $H_2O_2$. Three dimensional porous electrodes, such as RVC have high specific surface area, which directly influences in acceptable densities of electrical currents [10].

The RVC has been widely used as cathode in the electrofenton process. For instance, Fenton's reagent ($Fe^{2+}/H_2O_2$) has been successfully produced electrochemically under mildly acidic solutions using RVC cathodes, permitting the oxidation of 1 kg of phenol at $SnO_2$-coated Ti foam anodes with an associated electrical charge consumption of 6.3 kA h at 100 A $m^{-2}$ [11]. This was carried out by $Fe^{2+}$ ion addition or by reducing $Fe^{3+}$ ions electrochemically in the reactor with the simultaneous electrochemical production of $H_2O_2$ via reduction of dissolved oxygen. Discolouration and mineralization of azo dyes such as orange-G can be performed using the electrofenton reaction at $TiO_2$ anodes via the generation of $H_2O_2$ at RVC cathodes [12]. Furthermore, Levafix Blue CA and Levafix Red CA reactive dyes can be removed from waste waters by anodically electro-generated Fenton's reagent using RVC as cathode [13]. Additionally, Ghoneim et al., 2011 [14] and Vasconcelos et al., 2016 [15] have used electrofenton for the degradation of different dyes.

In some electrofenton systems, magnetite nanoparticles are added to the solution increasing the efficiency of degradation [16]. In these cases, after the process, the magnetite should be separated by magnetic harvest. To solve this fact, the magnetite nanoparticles can be integrated in the cathode electrode [17–19]. These nanoparticles are used as $Fe^{2+}$ source, activating the hydrogen peroxide to produce OH$^{\cdot}$ radicals through the Reaction (4).

$$Fe^{2+} + H_2O_2 \rightarrow OH^{\cdot} + OH^- \tag{4}$$

The novelty of this work focuses on the comparison of two materials (RVC and RVC/magnetite) as cathodes of an electrochemical reactor to produce $H_2O_2$ and the degradation of methylene blue (MB) as an organic pollutant. In addition, another difference with previous research is that no oxygen source will be used, which is challenging since the solubility of oxygen in water is low ($\approx 1 \times 10^{-3}$ mol dm$^{-3}$ at 25 °C) [20], and oxygen is critical for $H_2O_2$ production. In most published work, oxygen is supplied to the cathode surface by aeration or by using an external source of pure oxygen. We have opted for the use of an open system so we assume that the oxygen supply, although in low concentration, will be constant during the electrodegradation process. In addition, oxygen can be regenerated at the anode by decomposition of water according to the Reaction (5):

$$2H_2O \rightarrow O_2 + 4H^+ + 4e^- \tag{5}$$

## 2. Materials and Methods

### 2.1. Synthesis of Magnetic Materials on RVC Substrate

To synthesize magnetic nanoparticles on RVC, two different methodologies are carried out. First, with a procedure called A (dip coating) [21], consisting in the immersion of the RVC in a suspension of nanoparticles previously obtained by electrochemical synthesis [22]. Briefly, in the electrochemical synthesis the nanoparticles are obtained through the application of current to steel sheets arranged in parallel in a 3D-printed flow cell. The anodic reaction produces the oxidation of iron, while the cathode reaction produces the reduction of water resulting in the basification of the medium. Under these conditions the iron oxide nanoparticles are easily formed and then collected with a magnet. This suspension was prepared with 1 g/100 mL of nanoparticles and was vigorous stirring in ultrasound for 30 min. The RVC pieces of $4 \times 2 \times 0.8$ cm and 60 ppi was dip into the solution for the same time. The resulting RVC/coated were removed from the solution, washed several times and dried in an oven overnight at 80 °C. In the procedure B, the magnetic materials were obtained galvanostatically [23]. Briefly, a cell is initially filled with ultrapure water and degassed with $N_2$, the temperature is maintained at 80 °C in a thermostatic bath. Then 0.28 M $FeSO_4$, 0.026 M $NH_4$ and 0.057 M KOH are added to the cell. A constant anodic current of 100 mA is applied for 2 h, under a continuous flow of $N_2$. Finally, the electrode was removed, rinsed with plenty of water and vacuum drying at room temperature.

### 2.2. Characterization of the Electrodes

The morphology of the RVC and the RVC/coated was analyzed by Scanning Electron Microscopy (SEM) in a Hitachi S-3000N.

Phase identification and crystallographic structure of the magnetic materials deposited on RVC were analyzed by X-Ray diffraction (XRD) performed in a Bruker D8 powder diffractometer equipped with a primary monochromator and an ultrafast Lynxeye XE-T multichannel detector with $CuK_\alpha$ radiation. To carry out the measurements, the RVC electrode was dispersed in an agate mortar, taking the amount necessary for its analysis. The patterns were collected within 10° and 80° in 2θ, with an angular increment of 0.04° and increment time of 4 s. The diffractograms were analyzed using the PANalytical X'Pert HighScore program, and their profile fitting was carried out to obtain more reliable peak parameters.

To determine the ratio $Fe_3O_4/Fe_2O_3$, Mössbauer spectroscopy was used applying the called "center of gravity" method, which allows determining the ratio magnetite/maghemite nanoparticles from the area weighted mean isomer shift [24]. The spectra were measured at room temperature in triangular mode, using an emission spectrometer with a $^{57}Co/Rh$ source. Spectral analyses were performed by nonlinear fit, using the NORMOS [25] program, and energy calibrations were accomplished with α-Fe (6 μm) foil.

### 2.3. Degradation Processes

The degradation of methylene blue model compound was carried out in a glass reactor (200 mL) equipped with a magnetic stirrer (680 rpm). The initial concentration of methylene blue was of 20 mg $L^{-1}$ and 50 mM of $Na_2SO_4$. The electrolyte was stirred and maintained a stable pH (3.5) throughout the degradation process. The set up for the degradation is showed in Figure 1. A RVC or RVC/magnetic material is used as a cathode to reduce oxygen in solution to $H_2O_2$, and a graphite electrode as anode. Different currents are applied in a range of 25–200 mA. The MB electrodegradation efficiency, determined by UV-Visible spectrometry, was compared with the raw RVC.

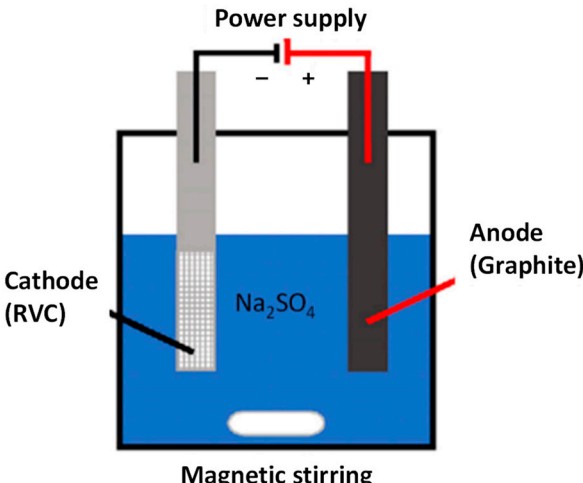

**Figure 1.** Schematic set up of the electrofenton system.

## 3. Results and Discussion

### 3.1. Characterization of the Electrodes

The morphology of the RVC used in this work can be seen in Figure 2a–c. We can observe the characteristic honeycomb structure of this material with a pore size of approximately 750 μm for a 60 ppi electrode (Figure 2a). Figure 2b shows this electrode after deposition of the nanoparticles by dipping according to procedure A. The formation of a compact layer of rough texture, with some aggregates of less than 1 μm can be observed. Figure 2c shows the galvanostatically coated RVC using procedure B. In this case, the formation of spherical nanoparticles, less than 1 μm, dispersed throughout the entire surface was observed. The formation of more compact coating using procedure A may be due to the heat treatment that was carried out after the deposition. An increase in the treatment temperature results in sintering of the nanoparticles.

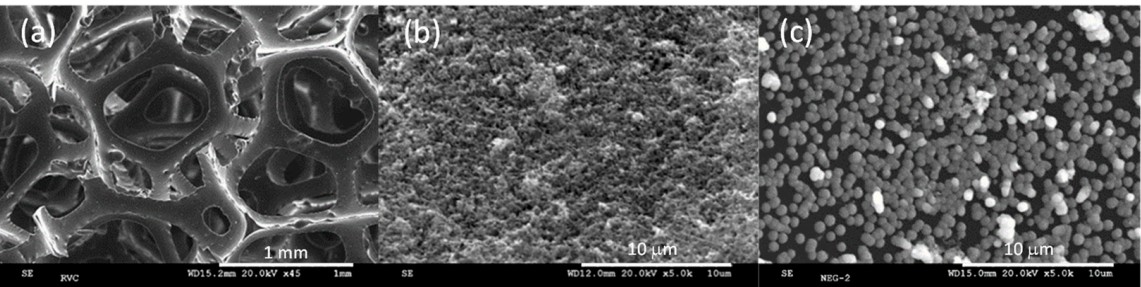

**Figure 2.** SEM images of studied cathodes: (**a**) 60 ppi raw RVC; (**b**) RVC dip coated; (**c**) RVC gal-vanostatically coated.

Figure 3a,b show the Mössbauer spectra for the RVC/coated samples of Figure 2b,c, respectively. The spectra show the superposition of the two sextets corresponding to the Fe in the tetrahedral and octahedral position of the spinel structure. The sample obtained by dipping (Figure 3a) has a weight mean isomer shift of 0.311 mm s$^{-1}$ derived from the spectrum fit, which corresponds to a 100% maghemite coating. On the contrary, the spectrum of Figure 3b that corresponds to the sample galvanostatically coated, shows a weight mean isomer shift value of 0.465 mm s$^{-1}$, which indicates 60% magnetite, the rest corresponding to maghemite. From the measurements made by Mössbauer it is impossible to determine whether there are two types of particles or whether there are magnetite particles with a certain surface oxidation.

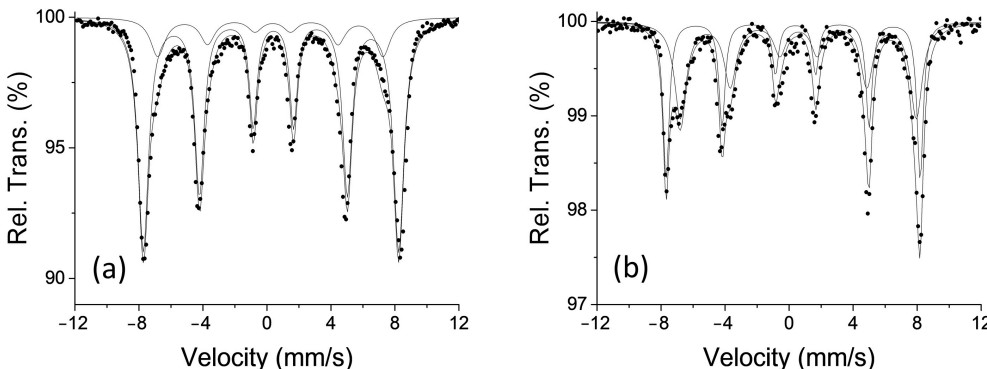

**Figure 3.** Mossbauer spectrum of coated RVC. (**a**) dip coated, (**b**) galvanostatically coated.

Figure 4 shows the diffractogram corresponding to the sample obtained using methodology A. All diffraction peaks match the standard JCPDS number 01-089-5892 which correspond to maghemite, in accordance with the Mössbauer results. There are also two reflections at angles of 25° and 22° 2θ, which corresponds to an amorphous phase of the RVC carbon electrode. The diffractogram confirms that there are no impurities corresponding to other iron oxides or oxyhydroxides. The diffractogram for the RVC coated galvanostatically (figure not showed) correspond to 100% magnetite (JCPDS 01-088-0315) and also all the peaks have been indexed to this phase of iron oxide.

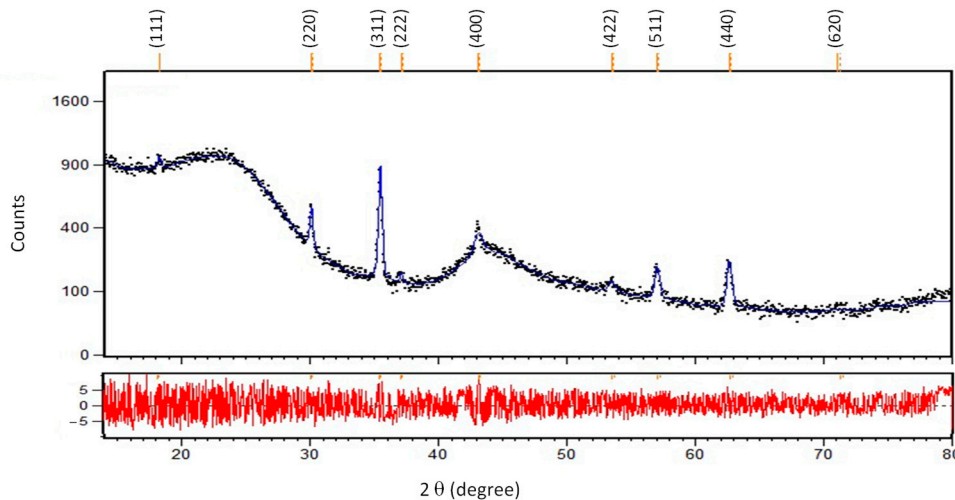

**Figure 4.** XRD diffractogram of dip coated RVC. The observed (black dots), calculated (blue line), and difference between fitted and observed patterns (red line) are shown. Bragg positions of the Fd-3m: 227 space group are included.

### 3.2. Degradation of Methylene Blue

Prior to performing the electrofenton process, the possibility of adsorption of MB on RVC and RVC sutrates with nanoparticles was verified. In none of the cases was an adsorption process observed. Mb, a cationic dye, is not adsorbed on nanoparticles because they are also positively charged at acidic pH levels.

Figure 5 shows the removal efficiency values of the MB degradation process using the raw RVC electrode, RVC coated with magnetic material according to procedure A and B, called RVC/A and RVC/B. It can be seen that the raw RVC produces a progressive discolouration of the medium up to reach 100% in about 100 min, applying a current of 100 mA. On the contrary, the degradation efficiency is higher for the RVC/A electrode, occurring 40 min after the application of current. In the case of the RVC/B electrode, in the first 15 min the degradation curve practically coincides with that of the RVC/A electrode,

but after this time there is a decrease in efficiency, and then after about 40 min it returns to increase. A possible explanation may be due to the low compaction of the particles on the RVC/B, as seen by SEM. At the beginning of the process, the $H_2O_2$ generated degrades the MB on the outer surface of the RVC/B, but in this case, on the inner part it will be slower due to the lower surface coating. When the MB near the surface is degraded its concentration decreases and therefore its efficiency, but due to a concentration gradient the MB will diffuse from the interior to the outer surface, increasing again the concentration and therefore the degradation efficiency. Since all the RVC/B electrodes used showed similar behavior, RVC/A electrodes were selected to accurately compare the effect of magnetic nanoparticles on MB degradation with uncoated RVC electrode.

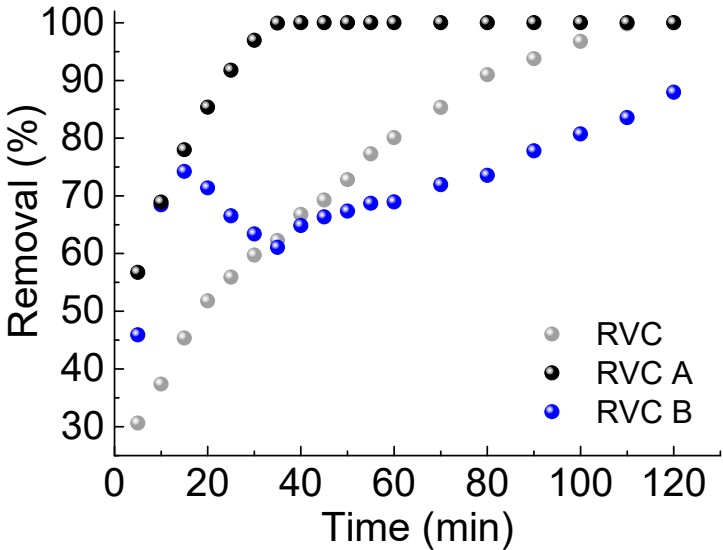

**Figure 5.** Efficiency in the discolouration of a 20 mg $L^{-1}$ MB solution by electrofenton, with applied currents of 100 mA.

Figure 6 shows the discolouration of 20 mg $L^{-1}$ MB solution using RVC dip coated at different applied currents. As can be seen to obtain a total degradation in a short time, currents above 100 mA must be used. These high current requirements can be explained by the methodology used. In general, the electrofenton process is limited by the low solubility of $O_2$ in the aqueous phase, as we have already commented. Furthermore, the efficiency of $O_2$ utilization is extremely low (<0.1%), and therefore a significant energy loss occurs, especially when the process is scaled. As the conditions in which we work are open cell and without forced oxygen passage, it is necessary to increase the applied current to be able to obtain a sufficient amount of peroxide. If the current is decreased the $H_2O_2$ production rate slows down and therefore the removal efficiency. In addition, the diffusive effects discussed above are observed.

The influence of temperature on degradation efficiency has been studied by applying an intermediate current of 50 mA (Figure 7). We can see how increasing the temperature from 25 to 50 °C produces an increase in removal efficiency. This increase may be due to an increase in ionic activity as well as a reduction in charge transfer resistance. However, when increasing to 70 °C there is a decrease in the efficiency of the process, which could be explained by a decrease in the solubility of oxygen in water [26] and therefore a lower production of $H_2O_2$, or by a decrease in $H_2O_2$ stability, both factors combined can produce a decrease in the amount of peroxide accessible for the degradation of MB.

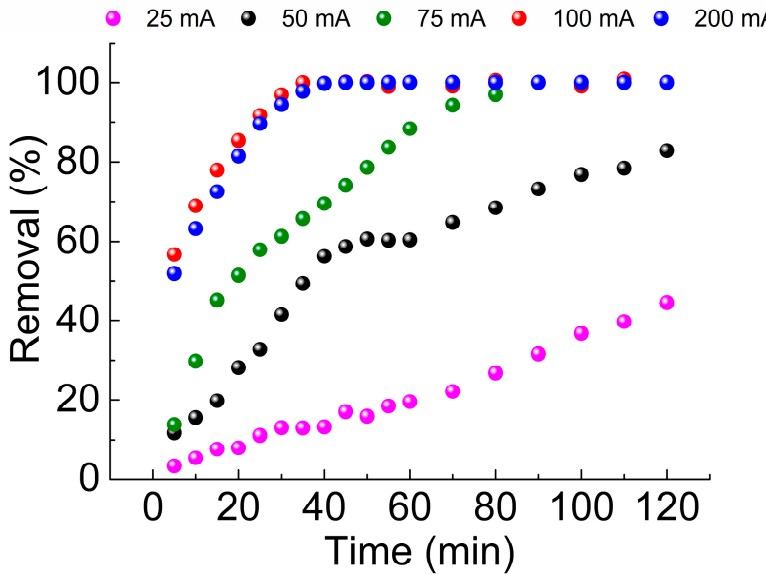

**Figure 6.** Effect of applied current on 20 mg·L$^{-1}$ MB degradation by electrofenton using RVC dip coated as cathode.

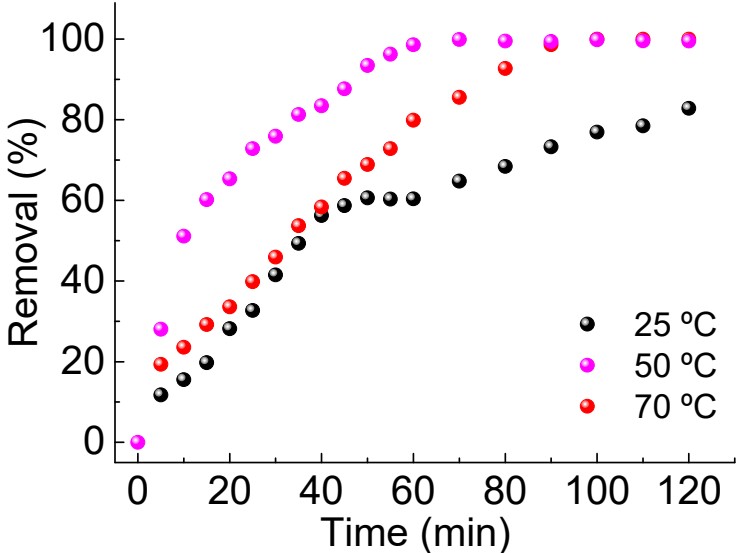

**Figure 7.** Influence of temperature on the electrofenton process (RVC/A). Applied current 50 mA.

The reproducibility of the process has also been studied, and the first step is to compare the reproducibility of different electrodes prepared by procedure A. Figure 8 shows the discolouration efficiency for five electrodes prepared under the same conditions. The amount of material deposited in each coating was calculated by weight difference between the uncoated and coated electrode, and the values obtained indicate a certain dispersion in the quantity of iron oxide nanoparticles deposited, which are between 17 and 30.3 mg. The time required for these electrodes under working conditions, to facilitate the total discolouration of the MB solution is about 47 ± 7 min, indicating that despite the dispersion in the amount of nanoparticles deposited on the RVC the fluctuation in the time for the total discolouration of the dye present a low deviation.

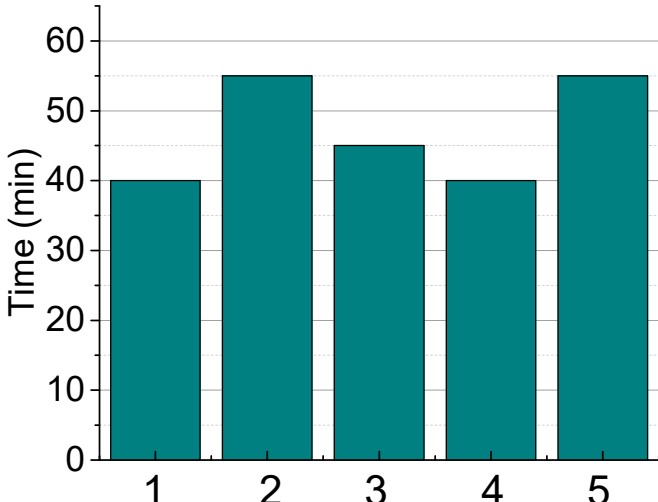

**Figure 8.** Time needed to reach 100% of MB removal using five different electrodes prepared with methodology A. T 25 °C, I = 100 mA.

It is also very important to know how many times we can use an electrode without losing its capacity and efficiency. Figure 9 shows the reuse of an electrode during four cycles. To carry out this experiment, the electrode was washed with abundant water after each process and dried in an oven overnight at 80 °C. Repeating the process in each cycle. The results shown a slight decrease in the degradation effectiveness in the second run, decreasing from 100% to 90%, but this value remains constant at least until the fourth cycle of electrode use.

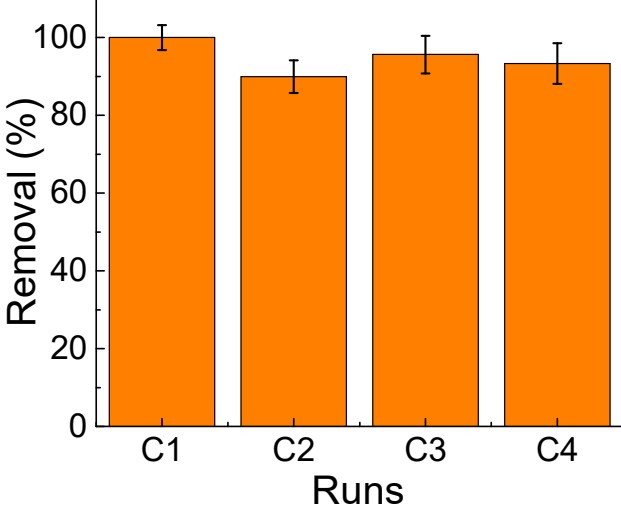

**Figure 9.** Reuse of a RVC electrode coated with magnetic nanoparticles using the A methodology, T = 25 °C, I = 100 mA.

## 4. Conclusions

In the current study, we have evaluated the use of two different methodologies for the deposition of nanoparticles on RVC electrodes. The dipping methodology results in a smoother deposit when compared to the galvanostatic methodology, which results in higher MB removal rates. The discolouration obtained by applying 100 mA and no extra oxygen addition that facilitates the peroxide formation is one of the highest reported. The dipping methodology is found reproducible, with a mean time to reach 100% discolouration of 47 ± 7 min when using different prepared electrodes. In addition, the as-deposited

NPs are attached well to the surface as they do not significantly decrease their degrading capacities within four reuse cycles.

**Author Contributions:** Conceptualization, P.H. and E.M.; methodology, F.L.R. and E.M.; formal analysis, P.H. and N.M.; investigation, F.L.R.; writing—original draft preparation, P.H., N.M. and E.M.; writing—review and editing, P.H., N.M. and E.M.; funding acquisition, P.H., N.M. and E.M. All authors have read and agreed to the published version of the manuscript.

**Funding:** This research was funded by Ministerio de Ciencia e Innovación, grant number PID2021-123431OB-I00. E. Mazarío. is grateful to the Madrid Government (Comunidad de Madrid, Spain) under the Multiannual Agreement with Universidad Autónoma de Madrid to encourage young research doctors within the context of the V PRICIT (Regional Program of Research and Technological Innovation, reference SI1-PJI-2019-00366).

**Institutional Review Board Statement:** Not applicable.

**Informed Consent Statement:** Not applicable.

**Data Availability Statement:** Not applicable.

**Conflicts of Interest:** The authors declare no conflict of interest.

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
