# Peer review of "Electrofenton with Reticular Vitreous Carbon and Iron Oxide Nanoparticles for Dye Removal: A Preliminary Study"

_applsci, doi:10.3390/app12168293_

Round 1

Reviewer 1 Report

The following comments to be reflected in the revised manuscript.

1. 3~4p : It is not clear how to determine the %removal of methylene blue. In general, a dye can be measured by its existence(%) using either a quantitative analysis, such as HPLC, or UV-VIS spectrophotometer.

2. What was the reason to choose methylene blue which belongs to a basic blue dye containing a cation ?

3. 3p : In the sentence of "~nanoparticles previously obtained by electrochemical synthesis [21]", the more descriptions are required for the nanoparticles.

4.  5~6p : It is required to propose the mechanism of electrochemical degradation of methylene blue.

5. Figure 5 : The difference of efficiency of two methods, RVC A & raw RVC, can be the time to decompose methylene blue. Therefore it is also required to provide more discussions about the role of nanoparticles based on the chemical properties.

Reviewer 2 Report

Dear authors,

The manuscript "Electrofenton with reticular vitreous carbon and iron oxide nanoparticles for dye removal" in general is well organized and written. Nevertheless, some key-points need to be reviewed and discussed with deeper fundamental arguments, as follows:

1. The reactions 1 to 3 could take place simultaneously. Therefore, discussion regarding the standard E value and thermodynamics must be provided to establish which one is preferred.

2. Explain why the sodium sulfate is exactly 50mM and what would happen if such concentration is different.

3. Figure 2. The increment of temperature could promote sintering of nanoparticles. Thus, what would be the effect on the effective area to produce hydrogen peroxide? A physisorption  characterization would give the area and pore dimensions that could be correlated with the electrode performance.

4. Figure 4. Please insert the JCPDS corresponding to magnetite and maghemite. In addition, what are the calculated and the observed diffractograms? It is not clearly indicated  in the Figure.

5. Figure 4. What is the explanation to have a not clear, sharp, well-defined peaks corresponding to (111), (222), (400), (422) and (620)?

6. Section 3.2.  Sorption study must be carried out previously to the electrofenton process, because the electrode structure allows diffusion toward its interior. Then, the methylene blue could be adsorbed and not necessarily it is degraded.

7. Figure 6. It is obvious that degradation kinetics is affected by current. What would be the explanation for the obtained results at 50, 75 and 100 mA, in terms of mechanism and kinetic?

8. Figure 7. You have results at three different temperatures. Then, you could obtain activation energy parameter and propose correlations with others results.

9. Page 7, lines 240-243. What the authors attribute such behavior to? If quantity of deposited material does not have influence in the performance, what is the mechanism of degradation?

10. The results must be discussed with deeper and fundamental arguments. Most of results are descriptive.

Reviewer 3 Report

In this study, the authors reported the degradation of methylene blue using iron oxide coated RVC electrode. Different iron oxides coated RVC electrodes were prepared and characterized in detail with, and the removal efficiency of methylene blue was measured. Satisfying performance for methylene blue degradation was achieved on the prepared electrodes. However, there are some revisions need to be made before acceptance for publication. Detailed comments:

(1)    Some grammar errors should be corrected.

(2)    Fig. 2, why the structure of RVC cannot be observed in b and c?

(3) Fig. 3, If possible, please provide Mössbauer parameters to specify the ratio of Fe3O4/γ-Fe2O3.

(3) The caption of Fig. 4 didn’t match what is described in the text. “galvanostatically coated RVC” was reported in the figure, while samples prepared unsing methodology A was described in the text. It is suggested that the XRD patterns of both materials should be provided to better support the results obtained on Mössbauer spectroscopy.

(5) Fig. 7, “T=25°C” in the caption should be revised. Actually, different temperatures were involved. In addition, the removal percentage of dye is higher than 100% at 50 degrees, why?

(6) Fig. 8, a bit hard to understand. Did the authors prepare 5 electrodes and test their performance? The result exceeds the range of 47 ± 5 min.

(7) What about the mechanisms for the reactions? I mean the dominant radicals or regent. Moreover, during the Fenton-like reactions, the release of Fe2+ and Fe3+ is key. More relevant information should be provided to improve the quality of the manuscript.  

(8) Did the material change after several runs of reaction?  

Round 2

Reviewer 2 Report

Dear authors,

Thank you for your point by point responses. If you consider that it is a kind of preliminary study to see the removal capacity by the electro-fenton methodology, then I suggest to change the title reflecting this. For instance: "Electrofenton with reticular vitreous carbon and iron oxide nanoparticles for dye removal: a preliminary study"

Author Response

Thanks to the referee for his/her suggestion. We think this is very appropiate and We have modified the title.